

# Prediction of cancer cell sensitivity to natural products based on genomic and chemical properties

Zhenyu Yue[1], Wenna Zhang[1], Yongming Lu[1],
Qiaoyue Yang[1], Qiuying Ding[1], Junfeng Xia[2] and Yan Chen[1]

[1] School of Life Sciences, Anhui University, Hefei, Anhui, China
[2] Institute of Health Sciences, Anhui University, Hefei, Anhui, China

## ABSTRACT

Natural products play a significant role in cancer chemotherapy. They are likely to provide many lead structures, which can be used as templates for the construction of novel drugs with enhanced antitumor activity. Traditional research approaches studied structure-activity relationship of natural products and obtained key structural properties, such as chemical bond or group, with the purpose of ascertaining their effect on a single cell line or a single tissue type. Here, for the first time, we develop a machine learning method to comprehensively predict natural products responses against a panel of cancer cell lines based on both the gene expression and the chemical properties of natural products. The results on two datasets, training set and independent test set, show that this proposed method yields significantly better prediction accuracy. In addition, we also demonstrate the predictive power of our proposed method by modeling the cancer cell sensitivity to two natural products, Curcumin and Resveratrol, which indicate that our method can effectively predict the response of cancer cell lines to these two natural products. Taken together, the method will facilitate the identification of natural products as cancer therapies and the development of precision medicine by linking the features of patient genomes to natural product sensitivity.

## INTRODUCTION

In recent years, many natural products were purified and shown to have cancer chemopreventive activity in laboratory, as exemplified by Camptothecin, Vinblastine, Embelin and Paclitaxel (*Dai et al., 2011*; *Goldwasser et al., 1995*; *Lynch et al., 2012*; *Of Trialists, 2011*). These agents from natural source have contributed significantly to the successful treatment of melanoma, leukemia, breast cancer and many other carcinomas. In addition, more and more new derivatives based on the structure of natural products have become promising candidates for antitumor drugs through laboratory design, synthesis and screening (*Chen et al., 2006*; *Rodríguez-Berna et al., 2014*; *Silvestri, 2013*). However, experimental methods for searching natural product lead structure suffered from the drawbacks of expensive and

Corresponding authors
Junfeng Xia, jfxia@ahu.edu.cn
Yan Chen,
chenyan91030@yahoo.com

time-consuming. Therefore the use of computational methods based on structure-activity relationship (SAR) has been intensively investigated.

Traditional exploring approaches of SAR focus on producing a range of analogues based on the basic skeleton of lead structure by synthetic chemists and searching empirically for their structural properties predictive of the antitumor activity (*Cao et al., 2013*; *Dong et al., 2012*; *Liu et al., 2015*; *Zhang et al., 2007*). These SAR studies tried to predict responses in a single cell line or a single tissue type using only structure data. Although much progress has been made, the problem of predicting natural products response is far from being solved.

In the present study, we developed a machine learning approach to predict the cell lines response to natural products, based on gene expression of cancer cell lines (genomic information) and the chemical descriptors of the considered natural products (chemical structure) for the first time. Empirical studies show that our method can obtain good performance when predicting sensitivity for hundreds of cancer cell lines to natural products in test set and case study analyses and indicate that both the structural properties and gene expression signatures are important determinants of antitumor activity of natural products. Taken together, this study outlines a first approach to predict drug response for natural products and generate novel natural product candidates for further studies.

## MATERIALS AND METHODS

### Data collection

In order to develop robust predictors of response to natural products, we collected and annotated a published large-scale preclinical dataset, namely, the Genomics of Drug Sensitivity in Cancer (GDSC) (*Garnett et al., 2012*). This large dataset includes drug sensitivity data from 138 drugs across almost 700 cell lines. By retrieving the drug information from PubChem database (http://pubchem.ncbi.nlm.nih.gov), we identified 17 drugs as natural products or their derivatives from these 138 drugs (Table 1). These natural products in GDSC were screened across a range of **279–565** cell lines per drug (mean = **495** cell lines per drug) representing **8,420** cancer cell line-drug interactions. The publically available drug sensitivity (Drug $IC_{50}$ values) data for all the 17 natural products was downloaded from GDSC (http://www.cancerrxgene.org).

Among these 17 natural products, 13 of them were randomly chosen for models' building, which represents 6,450 cancer cell line-natural product interactions (training set, Table S1). The remaining 4 natural products with 1,970 cancer cell line-natural product interactions were used in the test set (Table S2).

An independent test set (case studies) was extracted from the literature to further assess the performance of our proposed method. By searching anticancer herbs database of systems pharmacology (CancerHSP) (*Tao et al., 2015*) and natural products-related studies from the PubMed (http://www.ncbi.nih.gov/pubmed), we obtained two antitumor natural products (Curcumin and Resveratrol), which have been proven effective in inhibiting proliferation and inducing apoptosis of various kinds of cancer cell lines. For Curcumin,

**Table 1** Natural products and the corresponding cancer cell line-natural product interaction data used in the training set, test set and case studies analyses.

| Dataset | Natural product | Number of cancer cell line-natural product interaction |
|---|---|---|
| Training set | Vinblastine | 562 |
| | Parthenolide | 281 |
| | Rapamycin | 285 |
| | Thapsigargin | 559 |
| | Bleomycin | 559 |
| | Docetaxel | 562 |
| | Bryostatin 1 | 559 |
| | Cyclopamine | 279 |
| | Cytarabine | 562 |
| | Doxorubicin | 559 |
| | Embelin | 559 |
| | Mitomycin C | 559 |
| | Etoposide | 565 |
| Test set | Camptothecin | 562 |
| | Epothilone B | 559 |
| | Paclitaxel | 284 |
| | Shikonin | 565 |
| Case study | Curcumin | 7 |
| | Resveratrol | 8 |

it was screened on 16 cancer cell lines derived from 5 cancer types; and for Resveratrol, it encompasses drug sensitivity data for 13 cancer cell lines derived from 6 cancer types. After removing cell lines for which we could not find the corresponding gene expression information in GDSC, we finally obtained 7 and 8 cancer cell line-natural product interactions for Curcumin and Resveratrol, respectively (Table 2).

## Genomic features

The GDSC gene expression microarray data were derived directly from the work of *Geeleher, Cox & Huang (2014)*. Subsequent analyses were restricted to 12,026 annotated genes with Entrez Gene ID.

## Chemical features

The chemical features of the natural products were generated with PaDEL software (*Yap, 2011*) from the simplified molecular-input line entry system (SMILES) (*Weininger, 1988*). The SMILES files for natural products were collected manually from PubChem database (http://pubchem.ncbi.nlm.nih.gov). Initially, we obtained 1,444 1-D and 2-D descriptors of natural products directly from PaDEL. The chemical features with the same value across all natural products were further eliminated. Finally, we obtained 1,114 chemical features in this study.

**Table 2 Detailed results of case studies.** Samples, the number of cell lines in literature. Overlap, the number of cell lines overlapped with GDSC. Correctly, the number of cell lines whose sensitivity was predicted correctly.

**(A) Curcumin**

|  | Melanoma | Lung | Breast | Pancreas | Prostate | Total |
|---|---|---|---|---|---|---|
| Samples | 8 | 2 | 3 | 2 | 1 | 16 |
| Overlap | 4 | 1 | 1 | 1 | 0 | 7 |
| Correctly | 3 | 1 | 1 | 1 | 0 | 6 |

**(B) Resveratrol**

|  | Melanoma | Lung | Breast | Pancreas | Prostate | Neuroblastoma | Total |
|---|---|---|---|---|---|---|---|
| Samples | 2 | 1 | 3 | 2 | 4 | 1 | 13 |
| Overlap | 2 | 1 | 3 | 1 | 1 | 0 | 8 |
| Correctly | 1 | 0 | 3 | 0 | 1 | 0 | 5 |

## Models construction

In this study, the prediction model was built using the software WEKA (*Hall et al., 2009*) with the default parameters. The R scripts (*Ihaka & Gentleman, 1996*) were used for the statistical analyses.

# RESULTS AND DISCUSSION

## Strategy for prediction of cancer cell sensitivity to natural products

Our goal was to use gene expression and *in vitro* drug sensitivity data derived from cell lines, with the addition of chemical properties, to predict cell lines' response to natural products. The conceptual framework for prediction of cancer cell sensitivity to natural products is shown in Fig. 1. In the first step, cell lines in GDSC were clustered into two groups (Sensitive and Resistant) or three groups (Sensitive, Resistant and Intermediate) according to their sensitivities (drug $IC_{50}$ values) to a given drug with $K$-Means algorithm in WEKA (*Hall et al., 2009*). Here $K$ was set 2 or 3, which means that the cancer cell lines were divided into 2 or 3 groups. Samples in Sensitive and Resistant groups are used to build machine learning model. Then, the performance of J48 (Decision Tree), SVM (Support Vector Machine), Random Forest and Rotation Forest (*Rodriguez, Kuncheva & Alonso, 2006*) models were comprehensively evaluated. After this step, we used genomic features from gene expression data and chemical features to construct prediction model, where the optimal feature number were selected using $t$-test with R scripts (*Gentleman et al., 2011*).

## Determination of number of cancer cell lines clusters

To find the optimal number of cancer cell lines clusters in $K$-Means algorithm, the prediction performance of different clusters ($K$) were evaluated based on 10-fold cross validation (training set) and test set using Rotation Forest models. As can be seen in Fig. 2, the AUC (Area under the receiver operating characteristic curve) is higher in the case

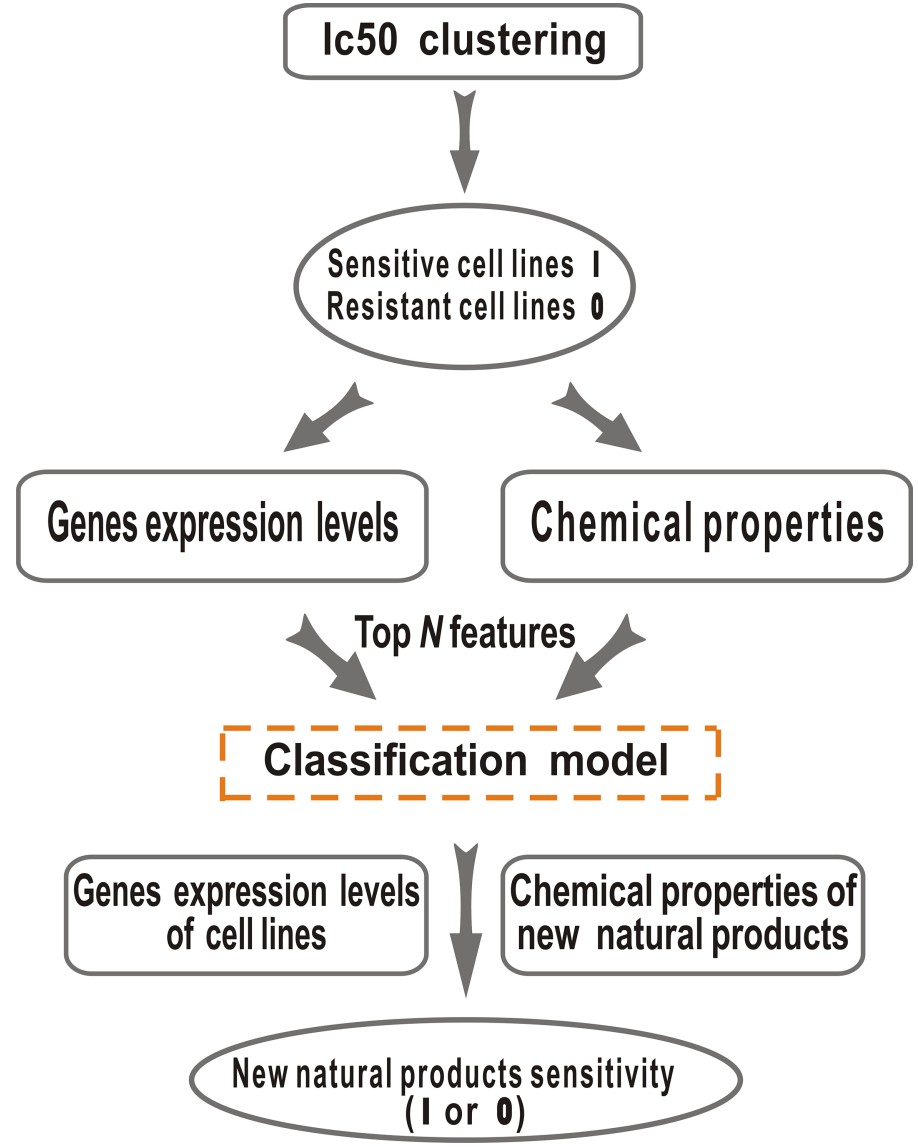

**Figure 1 Natural products sensitivity prediction workflow.** This data flow diagram demonstrates the simulation workflow for creation and optimization of natural products sensitivity prediction model. Our method was based on three key steps: (1) clustering of IC$_{50}$ values from the GDSC, then the cell lines were divided into 2 or 3 groups, the sensitive cell lines were set to 1, the resistant cell lines were set to 0. (2) Top $N$ features that were most significantly differential between the 1 and 0 cell line sets were chosen as the features of training and test sets. (3) Machine learning models were fitted in WEKA and can then be applied to the new data, to yield natural products sensitivity estimates.

$K = 3$ compared with those in the case $K = 2$ when features number is set as 50. The similar situation occurred when the features number is set as 100 or 500 (Figs. S1 and S2, respectively), so we chose $K = 3$, which means that the cancer cell lines in GDSC were clustered into three groups (Sensitive, Resistant and Intermediate), and only cell lines in Sensitive and Resistant groups were used in the subsequent analyses.

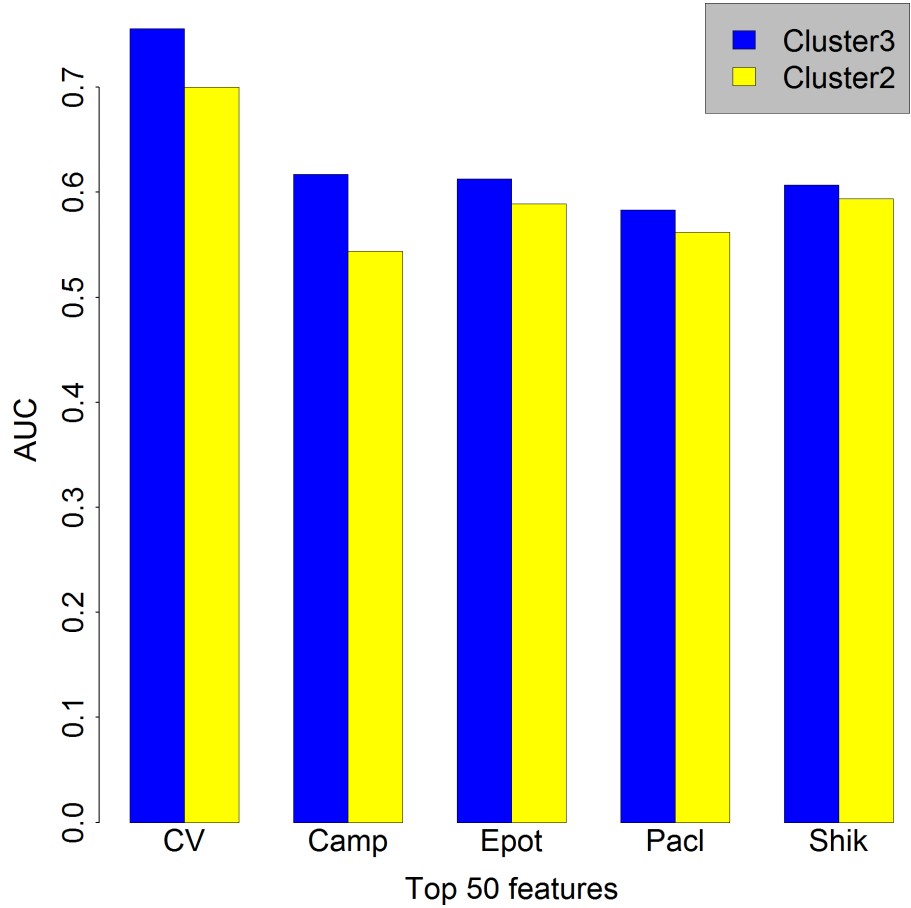

**Figure 2 Comparison between the case $K = 2$ and $K = 3$.** Bar chart showing in the case $K = 3$ (blue) we obtained a higher AUC than in the case $K = 2$ when features number is set as 50. Cluster3, the case $K = 3$. Cluster2, the case $K = 2$. CV, cross validation; Camp, Camptothecin; Epot, Epothilone B; Pacl, Paclitaxel; Shik, Shikonin; AUC, Area under the curve.

## Assessment of feature importance

In feature selection step, a 10-fold cross validation on the training set was conducted to get the optimal gene numbers. Examination on predicted AUC with respect to numbers of selected feature numbers showed a consistent trend of increasing first and decreasing afterwards with the increase of selected feature numbers except SVM model (Fig. 3). As a result, the top 1,000 features were chosen as optimal features for further analyses.

There were 468 genes (genomic features) in the top 1,000 features, of which 59 genes are cancer related genes (oncogenes or tumor suppressor genes), where oncogenes were obtained from database Cancer Gene Census (*Futreal et al., 2004*), and tumor suppressor genes were from database TSGene (*Zhao, Sun & Zhao, 2013*). We carried out a permutation test as follows. We randomly sampled 468 genes from the whole 12,026 genes 1,000 times, and the mean of the number of overlapped genes was only 36.2. In addition, the maximum value in the 1,000 times tests was 54, which is also less than 59. A *P*-value zero was obtained

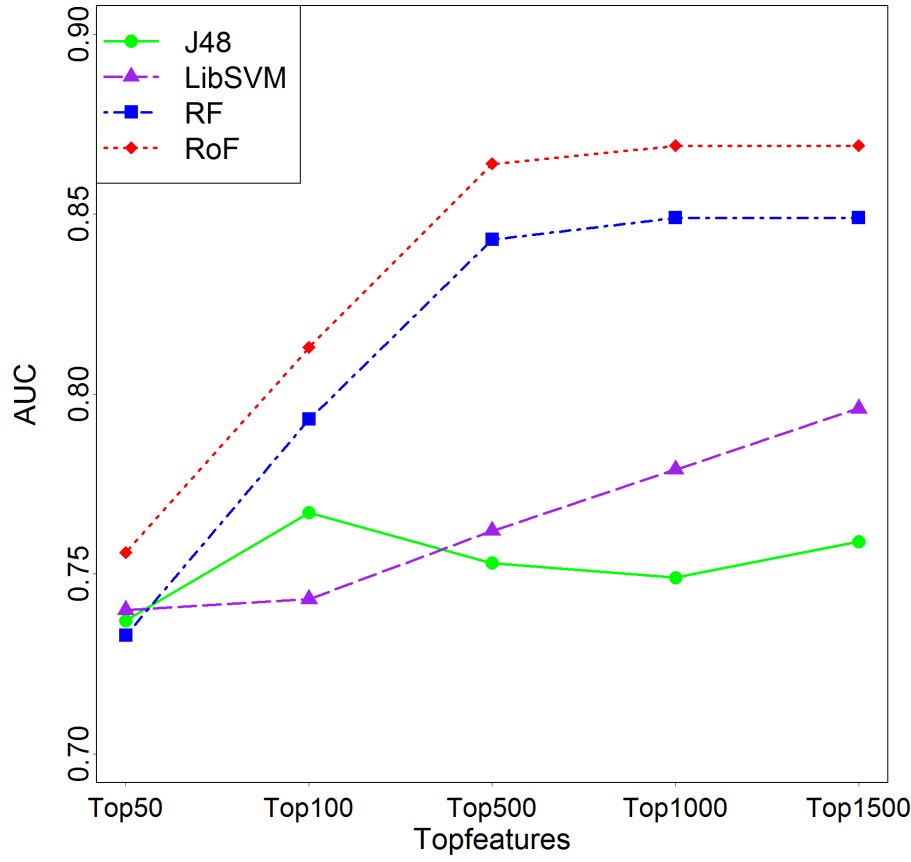

**Figure 3 Comparison among different top *N* significantly differential features.** AUCs plotted against different top *N* features in the case *K* = 3 are varied. J48, Decision Tree; SVM, Support Vector Machines; RF, Random Forest; RoF, Rotation Forest; AUC, Area under the curve.

in this permutation test. So the genomic features (genes) we chosen were more likely to be related to tumorigenesis.

The top 1,000 features also contained 532 chemical descriptors of natural products. The systematic machine learning-based integration of various data sources, including chemical structure and genomic information, can provide better discriminative power than those using only individual data sources. This may presents a simple and promising strategy to predict antitumor activity of unknown natural products using pharmacology data and machine learning approaches.

The detailed 468 genes and 532 descriptors used in the top 1,000 features are shown in Tables S3 and S4.

## Comparison of different machine learning methods

In this study, in order to identify the best machine learning technique suitable for predicting cancer cell sensitivity to natural products, we comprehensively evaluated the performances of SVM (LibSVM), Decision Tree (J48), Random Forest, and Rotation Forest classifiers. All these algorithms were implemented using the Weka package with the default parameter configuration. Rotation Forest has been proven to be a relatively stable machine

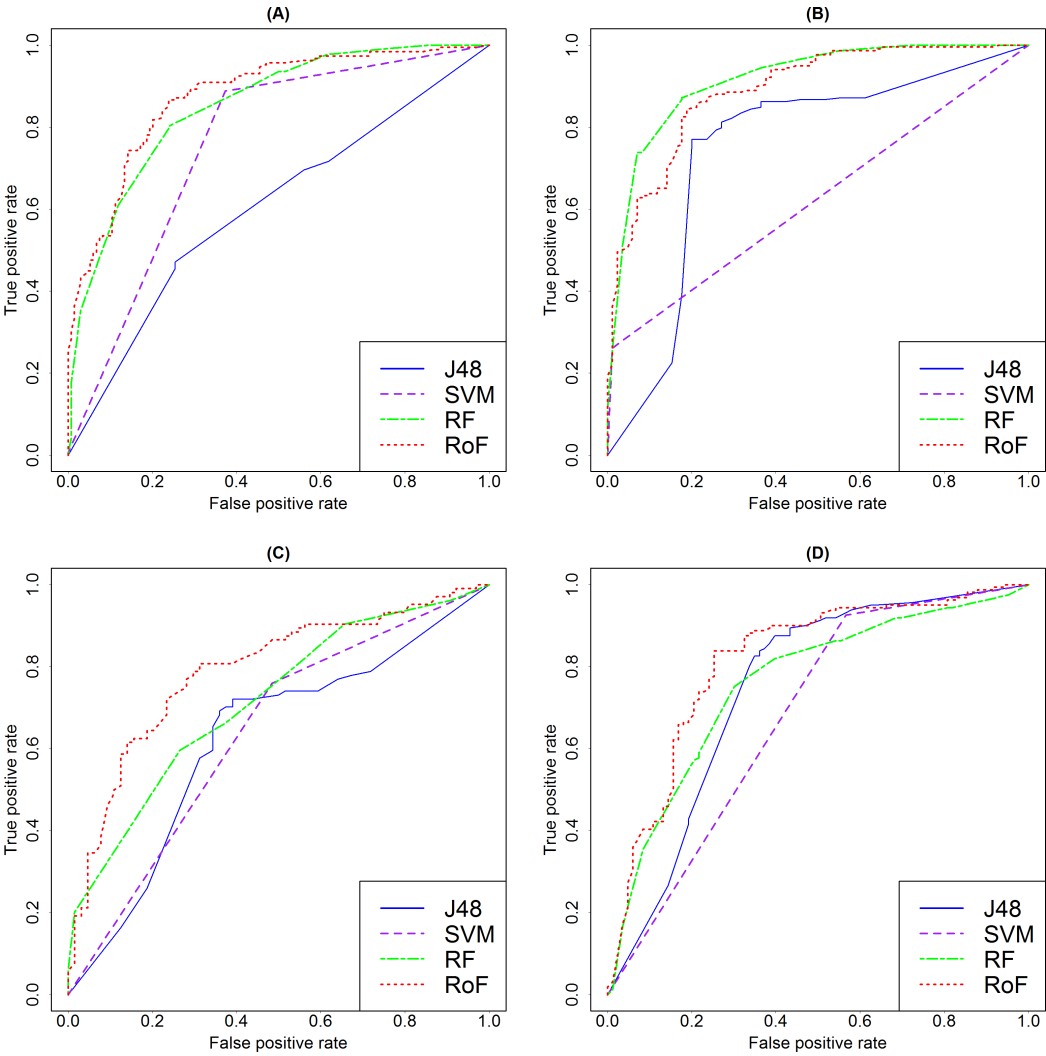

**Figure 4  Comparison among different machine learning models.** ROC curve showing the proportion of true positives against the proportion of false positives as the classification threshold is varied for test set. (A–D) represent Camptothecin, Epothilone B, Paclitaxel and Shikonin, respectively. ROC, receiver operating characteristic; J48, Decision Tree; SVM, Support Vector Machines; RF, Random Forest; RoF, Rotation Forest.

learning method in our previous work (*Xia, Han & Huang, 2010*), which also performed best using 10-fold cross validation (AUC = 0.87, Fig. 3) in this study. A consistent trend occurred in the test set (Fig. 4), where the AUC for for Camptothecin, Epothilone B, Paclitaxel, and Shikonin are 0.88, 0.89, 0.79 and 0.81, respectively.

The detailed classifiers assessment results of 10-fold cross validation (training set) and test set are shown in Table 3. The number of cancer cell line-natural product interactions for Camptothecin, Epothilone B, Paclitaxel and Shikonin are 321, 303, 168 and 244, respectively. The performance of each model is measured by five metrics: Precision, Recall, F-Measure, AUC and Accuracy (*Fawcett, 2006*), where Precision, Recall and F-Measure are calculated for each class, AUC and Accuracy are automatically weighted in WEKA for all

**Table 3  Results of classification based on different algorithms.** Detailed classifiers assessment results of training set and test set. Each dataset chose the top 1,000 group as the model features.

| Method | Class | Precision | Recall | F-measure | AUC | Accuracy |
|---|---|---|---|---|---|---|
| **(A) Cross validation** | | | | | | |
| J48 | 0 | 0.75 | 0.74 | 0.75 | 0.75 | 0.75 |
| | 1 | 0.76 | 0.77 | 0.76 | | |
| SVM | 0 | 0.77 | 0.78 | 0.77 | 0.78 | 0.78 |
| | 1 | 0.79 | 0.77 | 0.78 | | |
| RF | 0 | 0.75 | 0.83 | 0.79 | 0.85 | 0.78 |
| | 1 | 0.82 | 0.74 | 0.78 | | |
| RoF | 0 | 0.78 | 0.82 | 0.80 | **0.87** | **0.80** |
| | 1 | 0.82 | 0.78 | 0.80 | | |
| **(B) Camptothecin** | | | | | | |
| J48 | 0 | 0.50 | 0.75 | 0.60 | 0.61 | 0.58 |
| | 1 | 0.72 | 0.46 | 0.56 | | |
| SVM | 0 | 0.80 | 0.63 | 0.70 | 0.76 | 0.78 |
| | 1 | 0.77 | 0.90 | 0.82 | | |
| RF | 0 | 0.73 | 0.76 | 0.75 | 0.85 | 0.79 |
| | 1 | 0.82 | 0.80 | 0.81 | | |
| RoF | 0 | 0.84 | 0.69 | 0.76 | **0.88** | **0.82** |
| | 1 | 0.81 | 0.90 | 0.85 | | |
| **(C) Epothilone B** | | | | | | |
| J48 | 0 | 0.62 | 0.68 | 0.65 | 0.75 | 0.79 |
| | 1 | 0.87 | 0.84 | 0.85 | | |
| SVM | 0 | 0.34 | 0.99 | 0.51 | 0.63 | 0.47 |
| | 1 | 0.98 | 0.26 | 0.41 | | |
| RF | 0 | 0.58 | 0.92 | 0.71 | **0.92** | 0.79 |
| | 1 | 0.96 | 0.74 | 0.83 | | |
| RoF | 0 | 0.71 | 0.65 | 0.68 | 0.89 | **0.83** |
| | 1 | 0.87 | 0.90 | 0.88 | | |
| **(D) Paclitaxel** | | | | | | |
| J48 | 0 | 0.557 | 0.609 | 0.582 | 0.62 | 0.67 |
| | 1 | 0.745 | 0.702 | 0.723 | | |
| SVM | 0 | 0.569 | 0.516 | 0.541 | 0.64 | 0.67 |
| | 1 | 0.718 | 0.76 | 0.738 | | |
| RF | 0 | 0.533 | 0.625 | 0.576 | 0.72 | 0.65 |
| | 1 | 0.742 | 0.663 | 0.701 | | |
| RoF | 0 | 0.575 | 0.781 | 0.662 | **0.79** | **0.70** |
| | 1 | 0.827 | 0.644 | 0.724 | | |
| **(E) Shikonin** | | | | | | |
| J48 | 0 | 0.71 | 0.59 | 0.65 | 0.75 | 0.78 |
| | 1 | 0.81 | 0.88 | 0.84 | | |
| SVM | 0 | 0.75 | 0.43 | 0.55 | 0.68 | 0.76 |
| | 1 | 0.76 | 0.93 | 0.84 | | |
| RF | 0 | 0.59 | 0.70 | 0.64 | 0.76 | 0.73 |
| | 1 | 0.83 | 0.75 | 0.79 | | |
| RoF | 0 | 0.75 | 0.59 | 0.66 | **0.81** | **0.80** |
| | 1 | 0.81 | 0.90 | 0.85 | | |

**Notes.**

J48, Decision tree; SVM, Support vector machines; RF, Random forest; RoF, Rotation forest.

classes. As is shown in Table 3, all the 4 methods obtained good results based on 10-fold cross validation (training set) and test data set.

## Case studies

To further illustrate the effectiveness of our approach for detecting cancer cell sensitivity to natural products, we present two additional natural products examples. By searching CancerHSP database (*Tao et al., 2015*) and natural products-related studies from the PubMed database (http://www.ncbi.nih.gov/pubmed), we obtained 2 natural products screened on 29 cancer cell lines: Curcumin (*Bush et al., 2001*; *Choudhuri et al., 2002*; *Khor et al., 2006*; *Radhakrishna Pillai et al., 2004*; *Wang et al., 2006*) and Resveratrol (*Chen et al., 2004*; *Clément et al., 1998*; *Ding & Adrian, 2002*; *Hsieh & Wu, 1999*; *Lu & Serrero, 1999*; *Niles et al., 2003*; *Whyte et al., 2007*), which have been proven effective in prevention and treatment of various kinds of cancers, including melanoma, lung cancer, ovarian cancer and so on (*Tao et al., 2015*). After eliminating cancer cell lines for which we could not find the corresponding gene expression information in GDSC, we finally obtained 7 and 8 cancer cell line-natural product interactions for Curcumin and Resveratrol, respectively. The prediction results in these two natural products are shown in Table 2.

### Case study 1: curcumin

Curcumin, a phenolic compound from the rhizome of the plant *Curcuma longa*, induced apoptosis in tumor cells via a p53-dependent pathway or pathways independent of p53. We predicted responses of 7 cell lines that are sensitive to Curcumin, including 4 cell lines from melanoma, 1 cell line from lung cancer, 1 cell line from breast cancer, and 1 cell line from pancreatic cancer (Table S5). Notably, of the 7 cell lines that were defined as responders, 6 were correctly classified by our model (Table 2). The only cell line that was classified incorrectly is Sk-mel-5, a melanoma cell line containing wild-type p53. Because the rest 3 melanoma cell lines in this study contain mutant p53 (*Bush et al., 2001*), this may explain why our method could not obtain the correct result in Sk-mel-5 cell line.

### Case study 2: resveratrol

Resveratrol, a plant polyphenol found in grapes and a variety of human foods, is reported to have protective effects against various cancers. The mechanisms of its action in these diseases are inducing apoptosis via different pathways, antiestrogenic effect and so on. Responses of 8 cell lines to Resveratrol were predicted in this study, including 2 cell lines from melanoma, 1 cell line from lung cancer, 3 cell line from breast cancer, 1 cell line from pancreatic cancer and 1 cell line from prostate cancer (Table S5). SK-MEL-28, one of the two human melanoma cell lines used here, was predicted to be sensitive. The other melanoma cell line, A375, is amelanotic differing from the former. And Resveratrol induced phosphorylation of ERK1/2 in A375 which can promote gene expression associated with proliferation and differentiation, but not in SK-mel28 cells. Whether these differences contribute to the incorrect prediction of A375 cell line response to Resveratrol remains to be determined. Breast and prostate cell lines used here were all classified correctly. Altogether, 5 out of the 8 cancer cell line-natural product interactions can be correctly predicted by our model (Table 2).

## CONCLUSIONS

In this study, we investigated the inherent determinants of antitumor activity of natural products. For this purpose, we developed a machine learning method to predict natural products responses against a panel of cancer cell lines based on both the gene expression data and the chemical properties of natural products. Our results show that it is possible to enrich for natural products responders using gene expression and chemical descriptors, by applying models generated from a large panel of cancer cell lines. The performance of our approach was firstly evaluated using the 10-fold cross validation (training set) and test set, and further validated by modeling two additional natural products (case studies analyses). The experimental results show that our method can effectively predict the response of cancer cell lines to natural products.

Although our final best model is based on both the gene expression signatures of cancer cells lines and the chemical properties, novel features that better describe natural product sensitivity can be easily incorporated into our prediction system to further improve the prediction performance of natural product response. In our future work, we will add other genomic features such as mutation information into the prediction model. Besides these genomic information, epigenetic and protein level information also play very important role in natural product response mechanism, and thus should be incorporated in our prediction system. In addition, it should be noted that in the current study we focused on "natural product sensitivity in cancer." In the future, we will consider extending our model to non-natural product sensitivity prediction. Last, we will offer an online web interface through which our approach can be implemented to computationally predict natural product sensitivity.

### Funding

This work was supported by National Natural Science Foundation of China (31271817, 31301101), the Key Project of Science and Technology of Anhui (1501031099), the Anhui Provincial Natural Science Foundation (1408085QF106), the Specialized Research Fund for the Doctoral Program of Higher Education (20133401120011), and the Technology Foundation for Selected Overseas Chinese Scholars from Department of Human Resources and Social Security of Anhui Province (No. [2014]-243). The funders had no role in study design, data collection and analysis, decision to publish, or preparation of the manuscript.

### Grant Disclosures

The following grant information was disclosed by the authors:
National Natural Science Foundation of China: 31271817, 31301101.
Key Project of Science and Technology of Anhui: 1501031099.
Anhui Provincial Natural Science Foundation: 1408085QF106.
Specialized Research Fund for the Doctoral Program of Higher Education: 20133401120011.
Department of Human Resources and Social Security of Anhui Province: No. [2014]-243.

## Competing Interests

The authors declare there are no competing interests.

## Author Contributions

- Zhenyu Yue conceived and designed the experiments, performed the experiments, analyzed the data, contributed reagents/materials/analysis tools, wrote the paper, prepared figures and/or tables, reviewed drafts of the paper, data collection.
- Wenna Zhang, Yongming Lu, Qiaoyue Yang and Qiuying Ding reviewed drafts of the paper.
- Junfeng Xia conceived and designed the experiments, analyzed the data, contributed reagents/materials/analysis tools, wrote the paper, prepared figures and/or tables, reviewed drafts of the paper.
- Yan Chen conceived and designed the experiments, reviewed drafts of the paper.

## Data Availability

Sensitivity data for natural products/drugs:

http://www.cancerrxgene.org/

SMILES codes of natural products: http://pubchem.ncbi.nlm.nih.gov/

Gene expression of cell lines:

http://genemed.uchicago.edu/~pgeeleher/cgpPrediction/.

## Supplemental Information

Supplemental information for this article can be found online at http://dx.doi.org/10.7717/peerj.1425#supplemental-information.

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
