# Peer review of "Prediction of cancer cell sensitivity to natural products based on genomic and chemical properties"

_PeerJ, doi:10.7717/peerj.1425_

## Round 0.1 · original submission · Major Revisions

Both reviewers raise some important questions. Please prepare a point-by-point discussion and revise your paper correspondingly.

·

Basic reporting

The authors report their efforts to construct a model that predicts drugs sensitivity using gene expressions and chemical descriptors. It is my understanding that their analysis attempts to build a model of drug sensitivity using a training set of natural products (N=13), for 15 tumor cells that have gene expressions in the GDSC dataset and sensitivity measures to their set of natural products. The test set for this model is derived for 4 natural products, with the addition of curcumin and resveratrol. Apparently the intersection of their data, which is not clearly described, with the GDSC dataset yields these small numbers of cases (17 of their nautial products for 15 tumor cells). Restricting their analysis of only natural products severly will weaken claims that their approach is generally extendable to other cases. Once the reader understands this limitation, their conclusions must can be considered in this light. I would recommend that the authors strongly emphasize this limitation when proposing its extension to the broader area of non-narutal product drug sensitivity.

Experimental design

As noted above, the experimental design is narrowly focused on a small number of tumor cells and their drug sensitivity. Since their results are presented only within this set of data, the readers cannot assess the power of their analysis. In this regard, the opportunity to compare these results to responses in the remaining over 600 tumor cells, particularly with respect to gene responses. It in not clear whether this small set of gene responses is unique to the GDSC dataset. Gene clustering, as they have done with IC50's would provide some indication whether the 468 reporter genes is truely unique to these 15 tumor cell types.

The application of predictive models using, RF, SVM, J48 and RoF would be expected to yield relatively high measures of accuracy. Here they are using 468 genes expressions and ~1000 chemical descriptors to predict IC50 values of 13 tumor cells. Since standard t-tests were used to select these measures amongst all possibe measures (i.e. 12k gene expressions and numerous chemocal descriptors), one would have expected accuracies higher than ~75%.

Validity of the findings

From the context of their limited samping, their results are valid. Using ROCs across 4 different methods should not be widely different. All of these methods have some shared similarities in their calculations.

Additional comments

I believe your manuscript can be strengthened by inclusion of results that address the generality of this approach when applied to other compounds. Alternatively, the authors can restrict their claims only to this set of natural products.

Reviewer 2 ·

Basic reporting

No Comments

Experimental design

Validity of the findings

Additional comments

Zhenyu Yue and coworkers aim to develop a machine learning method to predict natural products responses against a panel of cancer cell lines based on both the gene expression and the chemical properties. This is an interesting work. However, these models are not fully verified by experiments. They would be better if minor modifications can be made. There is an anticancer herbs database (CancerHSP, http://lsp.nwsuaf.edu.cn/CancerHSP.php) where the authors can find data to verify the models.

---

## Round 0.2 · accepted · Accept

All comments have been addressed.